# Photoinduced multistage phase transitions in Ta₂NiSe₅

Q. M. Liu [1,5], D. Wu [1,2,5✉], Z. A. Li[3,5], L. Y. Shi[1], Z. X. Wang[1], S. J. Zhang [1], T. Lin[1], T. C. Hu[1], H. F. Tian[3], J. Q. Li[3], T. Dong[1] & N. L. Wang [1,4✉]

Ultrafast control of material physical properties represents a rapidly developing field in condensed matter physics. Yet, accessing the long-lived photoinduced electronic states is still in its early stages, especially with respect to an insulator to metal phase transition. Here, by combining transport measurement with ultrashort photoexcitation and coherent phonon spectroscopy, we report on photoinduced multistage phase transitions in Ta₂NiSe₅. Upon excitation by weak pulse intensity, the system is triggered to a short-lived state accompanied by a structural change. Further increasing the excitation intensity beyond a threshold, a photoinduced steady new state is achieved where the resistivity drops by more than four orders at temperature 50 K. This new state is thermally stable up to at least 350 K and exhibits a lattice structure different from any of the thermally accessible equilibrium states. Transmission electron microscopy reveals an in-chain Ta atom displacement in the photo-induced new structure phase. We also found that nano-sheet samples with the thickness less than the optical penetration depth are required for attaining a complete transition.

[1] International Center for Quantum Materials, School of Physics, Peking University, Beijing, China. [2] Songshan Lake Materials Laboratory, Dongguan, Guangdong, China. [3] Beijing National Laboratory for Condensed Matter Physics, Institute of Physics, Chinese Academy of Sciences, Beijing, China. [4] Collaborative Innovation Center of Quantum Matter, Beijing, China. [5]These authors contributed equally: Q. M. Liu, D. Wu, Z. A. Li. ✉email: wudong@sslab.org.cn; nlwang@pku.edu.cn

Ultrashort laser pulse not are only a powerful tool to excite and probe nonequilibrium electronic processes in transient states, but also emerges as an useful method to induce phase transitions that may or may not be thermally accessible. The latter has drawn increasing attention because it enables ultrafast optical manipulation and control over material properties[1–13]. Among various photoinduced (PI) phase transitions and related phenomena, the ultrafast switching from an insulating to a metallic state is particularly attractive, for it has high potential for device applications. While some phase transitions are transient, in the sense that they recover on very short time scales, e.g., picoseconds (ps), others are permanent or metastable and require something other than time to reset them. Up to now, the PI insulator to steady metallic state is achieved only in few systems exhibiting competing orders or broken symmetry phases, e.g., manganites perovskites[3,4,9,13] and a charge density wave (CDW) crystal of 1T-TaS$_2$[8], with their PI states only surviving at temperatures far below the room temperature. PI nonthermal insulator-metal phase transition was also reported in a strongly correlated material VO$_2$[14,15], however, a recent study questioned the claim and found no evidence for a hidden transient Mott–Hubbard nonthermal phase in VO$_2$[16]. Here we report on distinct PI multistage phase transitions with insulator-to-metal transition (IMT) characteristics in Ta$_2$NiSe$_5$—the compound is currently known as an important candidate of excitonic insulator (EI)[17–19]. We show that in Ta$_2$NiSe$_5$, the new steady phase induced by the strong laser pulses has a lattice structure different from any of the initial equilibrium states and is thermally stable up to at least 350 K.

## Results

Ta$_2$NiSe$_5$ is a two-dimensional (2D) van der Waals compound with typical in-plane quasi-one-dimensional (1D) substructure composed of parallel Ta and Ni atomic chains along a-axis of the lattice (Fig. 1a). Upon cooling, the system shows a second-order phase transition at $T_c = 326$ K accompanied by a slight shear distortion of those atomic 1D chains, that reduces the lattice symmetry from orthorhombic to monoclinic[19]. Our start point is the observation of the sharp resistance drop of ultrathin Ta$_2$NiSe$_5$ crystals induced by ultrafast laser pulses. The samples, with thicknesses of ~30 nm (less than the penetration depth ~55 nm at $\lambda = 800$ nm), are exfoliated from the bulk crystals (see Methods. Images of typical devices are shown in Supplementary Fig. 1). To

induce the resistance switching, laser pulses from a Ti:sapphire amplifier system with 800 nm wavelength, 35 fs duration and 1 kHz repetition were firstly used as the "writing" pulses with a fluence of ~3.5 mJ/cm$^2$. As shown in Fig. 1b, after writing pulses, the resistance drops approximately four orders of magnitude at 50 K thereafter retains in a low resistance (LR) state. This PI-LR state is thermally stable up to at least 350 K by the measurement. Unlike the pristine resistivity showing a transition at $T_c$ of 326 K, no anomaly is indicated in the temperature-dependent resistivity down to 1.8 K for PI-LR state (Supplementary Fig. 2). The activation energy of the PI-LR state is tiny and estimated as 4.3 meV, twenty times smaller than the one of pristine state. To induce the PI-LR state, a threshold fluence ~2.5 mJ/cm$^2$ is required. We remark that, due to the very high resistance at low temperature, we could not measure the value of the resistance for the pristine nano-thickness sample below 50 K. It is estimated that the resistance can drop by 8 orders at 4 K by such ultrafast laser pulse excitations. The PI-LR state is repeated from device to device, no matter the sample growth batches.

To gain insight into the microscopic nature of the PI-LR state, we investigated the time-resolved photoexcitations by pump–probe spectroscopy. In the experiments, the energies of pump and probe pulse are kept low at ~50 and ~3 μJ/cm$^2$ respectively, to ensure minimal disturbance of those states (see Methods). Figure 2a depicts the PI relative change of transient reflectivity $\Delta R/R(t)$. The rising time is about 100 fs. It can be clearly recognized that the decay process of PI reflectivity change $\Delta R/R(t)$ is superimposed by several coherent oscillations[20]. The reflectivity change could be well reproduced by two-exponential decay processes[21,22], a fast one in the time scale of 0.5–0.6 ps and a slow one with ~10 ps. Presence of rapid and slow decay dynamics after excitation has been observed in many systems. In general, the number of photoexcited hot electrons at zero time delay is related to the amplitude of $\Delta R/R$. Those excited high-energy hot electrons release and transfer their energy to lattice through the emission of optical phonons. The sub-picosecond decay would be mainly attributed to hot electron relaxation via the energy exchange with strongly coupled phonons, the ~10 ps decay process would be related to the energy exchange with other phonons or dephasing process. The fast decay process is in accordance well with the electronic pre-thermalization process revealed by tr-ARPES experiments[23,24].

The coherent oscillations arise from the coherent phonon excitations. We extract the coherent phonon spectra by fast

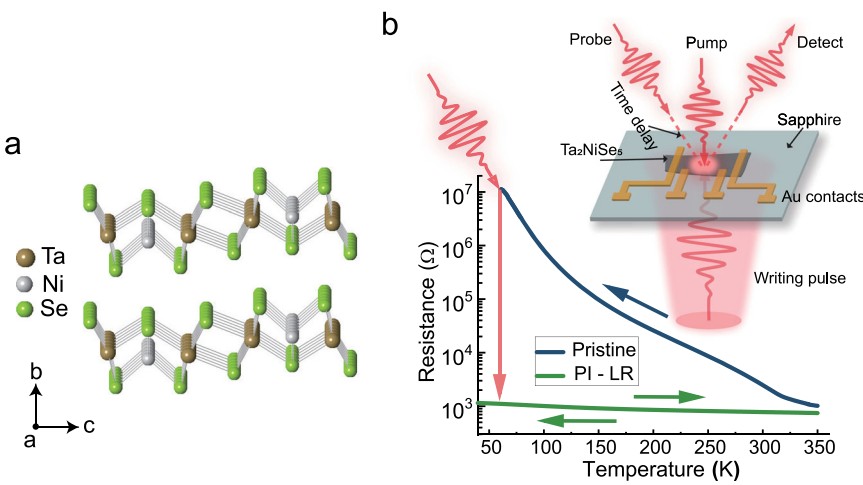

**Fig. 1 Resistivity switching of Ta$_2$NiSe$_5$ by the 35 fs laser pulse at 800 nm. a** The layered crystal structure of Ta$_2$NiSe$_5$. A quasi one-dimensional structure is formed by a single Ni chain and two Ta chains along a-axis. **b** The four-probe resistance of the pristine (blue) and PI-LR state (green) after a writing excitation ~3.5 mJ/cm$^2$. Inset is the schematic of the sample and experimental configuration. The length of dotted red line indicates the time delay between the pump and probe pulses.

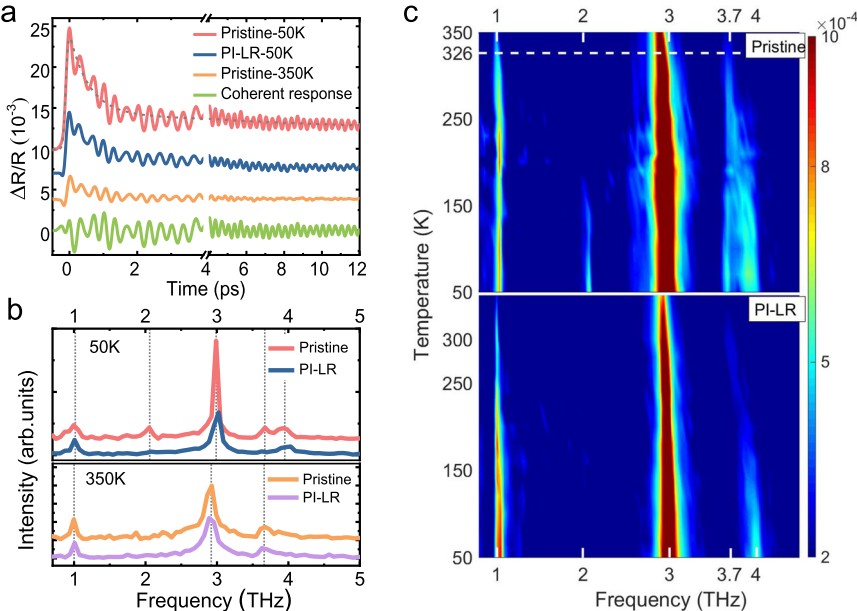

**Fig. 2 Pump–probe response spectra. a** Transient photoinduced reflectivity at various states. The signal is made up of the electronic response and the coherent oscillations. The gray lines represent fits to the measured data. **b** The corresponding FFTs from transient reflectivity spectra after substracting the incoherent components. **c** The temperature evolution of the coherent phonon spectra of pristine state (upper panel) and the PI-LR state (lower panel). In the experiment, the light polarizations of pump and probe are set parallel and perpendicular to a-axis respectively.

Fourier transformation from transient reflectivity spectra after subtracting the electronic decay background, as shown in Fig. 2b. For pristine state at 50 K, five phonon modes of 1.0-, 2.0-, 3.0-, 3.7-, 4.0-THz are detected. These modes are identified as $A_g$ symmetries in earlier studies[25], arising from displacive excitation of coherent phonons[26]. Remarkable differences can be found between the spectra of pristine state and PI-LR state at low temperatures. As shown in Fig. 2b, the 2.0- and 3.7-THz modes are absent for PI-LR state at 50 K. Besides, the peak frequencies of the remaining modes of PI-LR state are a little higher comparing with the ones of pristine state respectively. These changes in phonon number and frequency imply a PI lattice structural phase transition. More information of coherent modes evolution are recognized from the temperature-dependent spectra (Fig. 2c). For the spectra of pristine sample, the 2.0 THz mode is present solely in the low-temperature monoclinic phase, and the modes 3.7- and 4.0 THz gradually merge into one 3.7 THz across $T_c$ accompanying the structural transition from monoclinic to orthorhombic phase. While for the PI-LR state, the temperature-dependent spectra show only three recognizable modes throughout the measured temperature range, irrespective of thermal cycles. Although the PI-LR phase does not show a phase transition in the measured temperature range, the coherent phonon modes in the PI-LR phase are very similar to those in the high temperature pristine phase in $Ta_2NiSe_5$, therefore, it is expected that the PI-LR phase is in the similar lattice symmetry but with different atomic construction comparing to the pristine high temperature orthorhombic phase.

The structural transformation was directly clarified by transmission electron microscopy (TEM). Figure 3a depicts the perspective view of $Ta_2NiSe_5$ atomic model. An appropriate [110]-type zone-axis (Fig. 3b) of $Ta_2NiSe_5$ was selected for TEM structural characterization, by which atomically-resolved images can be readily obtained (see Methods). Figure 3c shows the electron diffraction pattern and the corresponding morphologies of a [110]-oriented pristine nanoflakes at room temperature. The stripe-like domain in TEM morphology and the splitting or

elongation of (3-30)-type reflections in diffraction pattern are arising from the twinning formation of monoclinic phase[27]. Upon heating across $T_c$, the lattice structure transforms into orthorhombic phase along with the disappearance of both the stripe-like domain and the splitting/elongation of (3-30) reflections (see Supplementary Fig. 3). The high-angle annual dark-field (HAADF) atomic imaging experiments were performed. Figure 3d shows an atomically-resolved HAADF image of the [110]-oriented pristine state at room temperature, in which the HAADF intensities scale with the atomic number of constituent elements: the brightest dots correspond to Ta ($Z = 73$) columns and the least bright dots correspond to Ni ($Z = 28$) columns. A periodic structural unit is framed and false-colored in Fig. 3d, and its enlargement is shown in Fig. 3e together with the corresponding positions of Ta columns in the [110] projection. Note that it is hard to differentiate the HAADF atomic images between the low-temperature monoclinic and high temperature orthorhombic phases of pristine $Ta_2NiSe_5$ in the limitation of TEM experiments, consistent with the fact that the two pristine phases have quite tiny differences in crystallographic parameters as revealed by X-ray experiments[28].

Similar TEM structural characterizations were carried out for writing-pulse excited $Ta_2NiSe_5$ nanoflakes. Figure 3f shows the TEM morphology and the corresponding electron diffraction of [110]-oriented $Ta_2NiSe_5$ after 800 nm pulses (at ~3.5 mJ/cm²). Clearly, the twinning related stripe contrast in TEM morphology and the splitting of reflections in diffraction pattern are both absent, in stark contrast to the results (Fig. 3c) for the pristine state (see Supplementary Fig. 4 for more [010]-oriented). Figure 3g shows a representative [110]-oriented atomically-resolved HAADF image and the Fig. 3h is the enlargement of the periodic structural unit frame correspondingly together with the Ta-columns extracted in [110]-projection. The clean and uniform atomic periods in Fig. 3g suggest that there no observable PI vacancies or defects in the induced hidden state. Despite the greater similarity in the chain-like structure along a-axis between Fig. 3d, g careful inspection of their respective atomic

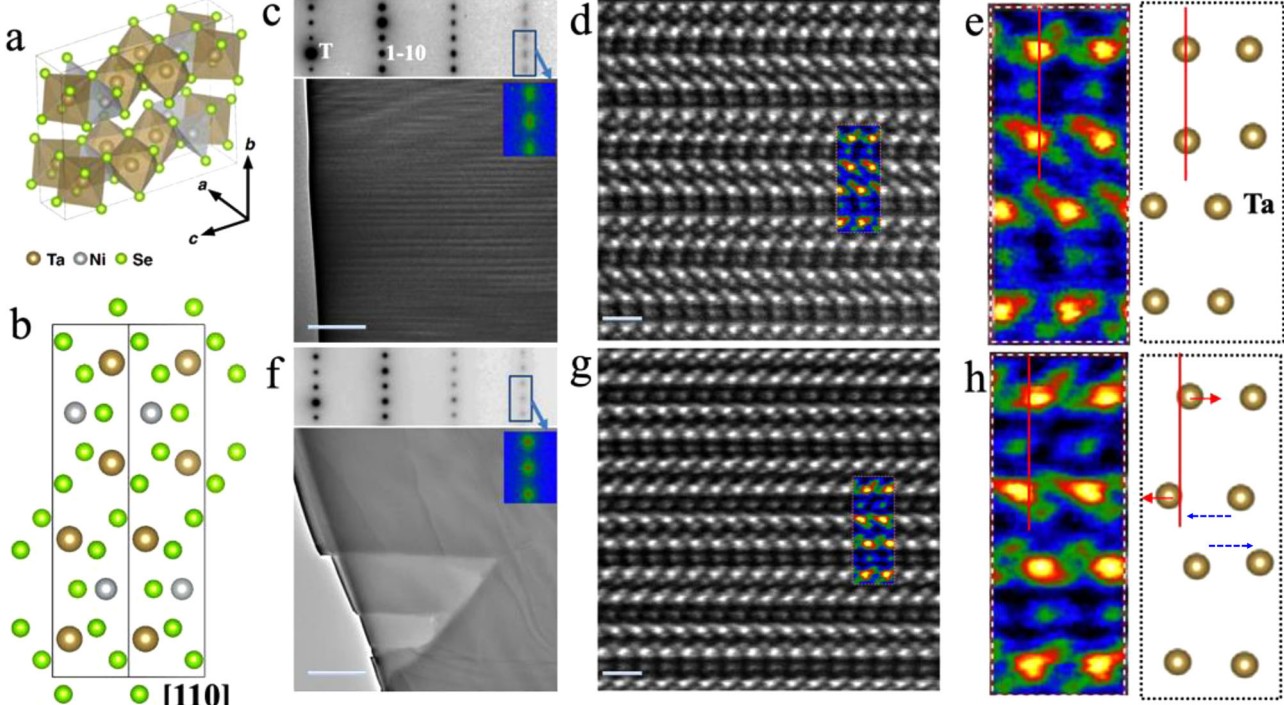

**Fig. 3 TEM structural characterization of both pristine and laser-treated Ta$_2$NiSe$_5$. a** Perspective view of Ta$_2$NiSe$_5$ atomic model. **b** Projection view along [110] zone-axis. **c** For pristine Ta$_2$NiSe$_5$, [110]-oriented electron diffraction pattern and morphology, inset shows the zoom-in part of the splitting of (3-30) reflections due to the twinning formation. **d** Atomically-resolved high-angle annual dark-field image taken along [110] zone-axis corresponding to (**c**). **e** Enlarged part outline in (**d**), with a schematic of Ta atom columns corresponding to the brightest contrast. Descriptions for (**f, g, h**) are the same as (**c–e**) but for hidden state Ta$_2$NiSe$_5$. Red line and arrows in (**h**) mark the small relative displacement of Ta atoms with respect to the one in (**e**). The blue dashed arrows denote the locate where large Ta-atoms relative displacement occurred. Scale bars in (**c, f**) correspond to 500 nm, in (**d, g**) correspond to 0.5 nm. False colors scales with the image intensities for the purpose of visual clarity.

arrangements unveils notable Ta-lattice shear distortion comparing Fig. 3e, h. The colored lines and arrows in both Fig. 3e, h mark the Ta lattice displacement, we can recognize clearly a small and a large shear motion respectively between different Ta-chains. The small one can be considered as a shear motion of the Ta against the Ni chains, with a counter Ta-Ta displacement measured to be about $0.5 \pm (0.05)$ Å along <110>-type direction for the PI-LR phase. While the large shear motion recorded a relative displacement of about $1.6 \pm (0.05)$ Å along <110>-type direction, which occurs between the two nearest-neighboring Ta-chains, where the weakest Ta-Se bonds exist. Considering the large shear motion of Ta lattice, geometrically, it corresponds closely to a $a/2$ $a$-axis sliding between the nearest-neighboring Ta-chains (see Supplementary Fig. 5 for details).

It would be interesting to compare the above coherent phonon excitations and structural characterization with the equilibrium structural transition revealed by Raman scattering studies[29–32]. Raman studies have revealed significant role played by the 2-THz mode in the structural phase transition. The mode has a B$_{2g}$ symmetry in the high temperature orthorhombic phase but evolves into an A$_g$ symmetry in the low-temperature monoclinic phase. It corresponds to the in-chain Ta lattice oscillations or shear motion of TaSe$_6$ octahedra along the a-axis[29]. An instability of this mode as the origin of phase transition was suggested by density function theory based calculations[29] and indeed observed in a recent Raman measurement[30]. In the present study, the optical pump–probe experiment detects only the A$_g$ coherent phonon modes. In the pristine state below $T_c$, five A$_g$ modes were detected at 50 K and two of them (2.0- and 4.0 THz) disappeared due to evolving into B$_{2g}$ modes above $T_c$, the result is in

accordance with the Raman measurement. On the other hand, the 2-THz mode is also absent in the PI-LR phase. In fact, the pump fluence dependent pump–probe measurement in the monoclinic phase presented below indicates a clear weakening and disappearance of the 2-THz mode upon increasing fluence. It is expected that an offset displacement gradually develops in the direction of the in-chain phonon mode, and the oscillatory motion would be on top of that. Increasing the pump field would lead to a much larger offset displacement, similar to what has been observed in WTe2 and MoTe2[33,34]. Above a threshold pump field, the lattice would not recover to its original state and a new equilibrium state is reached eventually. The in-chain Ta atoms displacement found by TEM agrees well with the lattice instability of B$_{2g}$ mode proposed by the Raman work. Nevertheless, the large displacement of Ta atoms revealed by TEM measurement yields evidence that the PI new state is different from any of already known phases in pristine Ta$_2$NiSe$_5$. As the Ta lattice sliding has an intimate correlation with the EI state in Ta$_2$NiSe$_5$[30,31], it can be expected that the pristine exciton condensation will be destroyed as soon as the Ta lattice transformation occurred. Note that previous report has revealed a high-pressure induced $a/2$ interlayer-sliding transition in Ta$_2$NiSe$_5$ with the lattice transformed from monoclinic to orthorhombic symmetry along with a IMTs electronic transition[35]. As well, in pristine Ta$_2$NiSe$_5$, the electronic transition at 326 K is accompanied concomitantly with a lattice transition with the Ta sublattice slightly sheared along a-axis. It is therefore not surprising that the Ta lattice transformation will come along with an electronic phase transition in our observations in Ta$_2$NiSe$_5$ system.

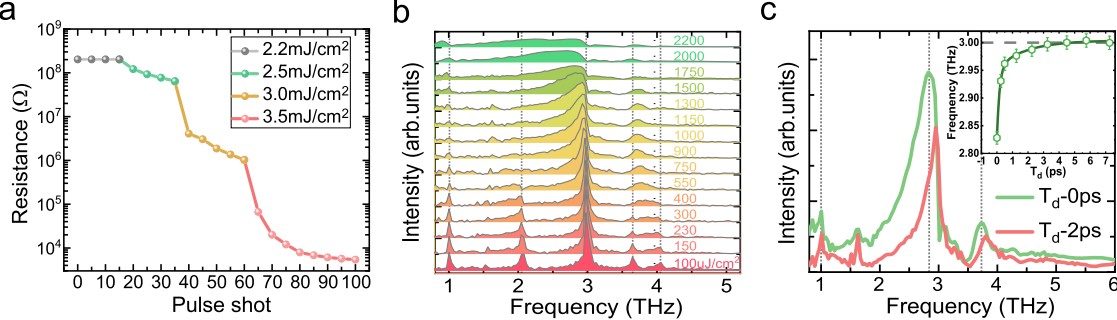

**Fig. 4 Pulse fluence threshold and Fluence dependent PI state properties. a** Shot-to-shot resistivity change of $Ta_2NiSe_5$ ultrathin sample at 50 K. Each shot includes a sequence of five pulses. **b** Fluence dependent coherent phonon spectra for moderate fluences at 50 K. The energy of probe pulse kept constant in a small value of ~3 μJ/$cm^2$. The dashed are guidelines. **c** The red-shift effect of the 3.0 THz mode as a function of the time delay $T_d$ for incident 1.5 mJ/$cm^2$. The inset shows an exponential approximation recovery to the initial frequency, where the error bar is the standard deviation evaluated when extracting the peak frequency from the coherent phonon spectra.

Recent structure investigation combined with laser-pumping and first-principle calculation study on layered van der Waals CDW material 1T-$TaS_2$ suggested that the interlayer electronic ordering can play an important role in electronic phase transitions[36,37]. The metal-insulator transition is driven not by the 2D order itself, but by the vertical ordering or stacking of the 2D CDWs. Among 13 different stackings, there exist two exceptionally stable stacking configurations, one being an insulator and the other being a metal. They have a small energy difference, and their competition is responsible for the metal-insulator transition in 1T-$TaS_2$[37]. The stacking scenario was used also to explain PI insulator-metal phase transition in 1T-$TaS_2$ compound. However, the observed in-chain displacement of Ta-Ta atoms in $Ta_2NiSe_5$ in the PI LR state indicated that the distortion appears more complicated than a simple stacking fault proposed in ref. [37].

To generate the PI-LR state in $Ta_2NiSe_5$, a pulse fluence beyond a threshold is required. Figure 4a shows a shot-by-shot dependence of resistance measured at 50 K. It records a steep drop of resistivity following the excitations beyond a threshold fluence. From the figure, the threshold is estimated near ~2.5 mJ/$cm^2$. Each subsequent shot will drive the resistance drop into a new steady step value until another shot incident. With the shot number increasing, the resistivity drops quasi-exponentially to a local minimum value. Higher density excitations will drive the resistance to smaller values till reaching a minimum constant. This step-by-step type behavior is closely coincident with the phase percolation scenario, as it was found in transition metal oxide $La_{2/3}Ca_{1/3}MnO_3$[9]. That means the resistance change does not simply arise from the number of absorbed photons. The writing pulses can induce a range of resistivity values, indicating the multiphase coexistence of pristine and PI-LR phases, dependent on the shot number and intensity. Thus, we can imagine an exponential growth of the excited volume fraction of PI-LR phase with increasing shot number or intensity. We also found that the same PI-LR phase can be driven by laser pulses with longer pulse duration of 100 fs but at higher fluence, suggesting that the pulse field plays a dominant role in the PI effect. We remark that, to achieve the PI-LR state, the delicate substrate environment has to be taken into account. In $Ta_2NiSe_5$, the PI-LR state is always detected when the sample's thickness is thinner than the light penetration depth. However, for thicker bulk samples the 2.0- and 3.7-THz modes are usually recovered after the writing pulses ceased (see Supplementary Fig. 6), with a high resistance state being kept as well. The observations are understandable. For the exfoliated thin flake sample transferred on the substrate with thickness smaller than the optical penetration depth, once the PI-LR state is triggered, the underlying lattice would be completely transformed into a new phase that is weakly affected from the substrate. Whereas for a thick bulk sample, the underneath unexcited or incompletely transited lattice may inevitably exert a restoring strain to the upper excited layers, resulting in a possible recovery of the 2.0- and 3.7-THz modes when the pump fluence is below the damage threshold. Switching of electronic properties more easily in thinner superlattice samples by femtosecond laser pulse has also been reported in other phase change materials, for example, in $Ge_2Sb_2Te_5$ in terms of strain effect[38].

For relative weak pulse energies, the temporal photoresponse spectra behave in a different rule. Figure 4b presents the coherent phonon spectra in a pristine nano-thickness sample with an increase of pump fluence up to 2.2 mJ/$cm^2$. The original oscillations in the time domain is presented in Supplementary Fig. 7. Initially at weak pump fluence, the primitive five coherent modes are recorded. With the pump fluence increasing, the 2.0 THz mode tends degraded gradually and disappear completely above ~550 μJ/$cm^2$, while the 3.7- and 4.0-THz modes get merged into a broad one, suggesting a PI structural phase transition. However, this transition is transient, it doesn't yield a permanent reduction of DC resistivity and the coherent phonon spectra will recover to the initial state when the pump fluence is tuned back to small values (see Supplementary Fig. 8). With the excitation intensity increasing, the three low frequency modes show a clear red shift and an asymmetric broadening to the lower-frequency side, the phenomena can be ascribed to coherent anharmonic oscillations under high density photoexcitations[39,40]. The red shift of the peak frequency can be considered as an electronic softening effect relating with the transient photoexcited carriers. While for the asymmetric broadening modes, the effect is typically referred as the nonthermal modification to the lattice potential-energy surface at high-intensity excitations[41,42]. This broadening modification is a transient effect that only lasts for a short time before thermal equilibrium is attained between electrons and lattice background. We performed the time-domain trace analysis for the modification effect. Figure 4c shows the Fourier transformed spectra obtained after cutting the initial time delay $T_d$ at two different positions for a pump fluence 1.5 mJ/$cm^2$. As can be seen, if we perform Fourier transformation of the pump–probe waveform after the first 2 ps ($T_d = 2$ ps), the red shift and broadening of the phonon modes approach to disappear. The inset shows the change of 3 THz mode of Fourier transformed spectra after cutting different time delays $T_d$. A more detailed analysis of the transient phonon spectra as a function of cutting

time delay $T_d$ at various excitation intensity is presented in Supplementary Fig. 9.

## Discussion

We now discuss implications of the results. First, our experiments clearly demonstrate multi-staged PI phase transitions. At small excitation fluence, the system is triggered to a transient or short-lived state accompanied by a structural change. Upon increasing the pump fluence beyond a threshold, the sample is driven to a stable phase showing high conductivity. This phase has a crystal structure different from the original one with observable sliding of Ta-atom, which is also robust against thermal excitations up to at least 350 K. Obviously, this phase must correspond to one of the pronounced minima in the free energy landscape. The energy barrier is high enough to prevent the system from thermally recovering to the pristine phase. In principle, the PI atomic vacancies might provide such a stabilized energy barrier, like a case of irreversible IMTs from insulating 2H-MoTe$_2$ to metallic 1T′-MoTe$_2$ induced by high-intensity photo-irradiations[43]. However, this possibility can be excluded here in Ta$_2$NiSe$_5$ system, because our atomically-resolved TEM experiments indicate a homogeneous atom arrangement with negligible vacancy or disorder in the PI state of Ta$_2$NiSe$_5$. Therefore, the PI multistage phase transitions in Ta$_2$NiSe$_5$ must be a result of self-organization through different trajectories from photoexcited states.

Second, our measurements provide useful information about exciton formation in Ta$_2$NiSe$_5$ system. The binding energy of exciton appears to be very sensitive to the lattice distortion. The phase of Ta$_2$NiSe$_5$ below $T_c = 326$ K has been widely considered as a prototype EI[17–19,44–46]. As is known, for a small gap semiconductor, an excitonic instability occurs only if the exciton binding energy is larger than the electronic band energy gap. Then, the system would form a new phase through spontaneous formation and Bose condensation of excitons in which the exciton binding energy would become smaller than the band energy gap. Our measurement reveals the presence of a nonthermal stable phase with subtle difference in structure relative to the pristine Ta$_2$NiSe$_5$.

The absence of a phase transition suggests very small exciton binding energy in this phase. Therefore, the subtle structural change could induce a large change in the exciton binding energy. Nevertheless, the driving mechanism for the PI phase transitions may not be directly linked to the exciton interaction. Although it is not easy to solve the issue conclusively at present stage, a comparative study to the pure semiconductor Ta$_2$NiS$_5$, an isostructural sister compound to Ta$_2$NiSe$_5$ without exhibiting exciton insulator ground state[17], would be particularly helpful. To this end, we performed similar measurement on Ta$_2$NiS$_5$. Indeed, we observed similar PI-LR state in Ta$_2$NiS$_5$ (see Supplementary Fig. 10). The results indicate that the exciton interaction is not a prerequisite for the formation of PI-LR state.

Finally, our work reveals a significantly different PI effect from reported work for the compound. The issue of PI phase transition in Ta$_2$NiSe$_5$ has been heavily debated with controversial results. Several reports denied a possible of PI phase transition[23,47,48], however some recent reports support that transition[24,25,49,50]. For example, in earlier tr-ARPES and ultrafast pump–probe studies on Ta$_2$NiSe$_5$[23,47], Mor et al. reported that the photon absorption saturated at a critical fluence of $F_c = 0.2$ mJ/cm$^2$ upon photoexcitation, and above the saturation fluence the band energy gaps widened, reflecting an increase of the exciton condensate density in the EI phase in Ta$_2$NiSe$_5$[23]. In accord with the tr-ARPES results, their pump–probe measurement also indicated a saturation phenomenon above a critical excitation fluence[47]. Based on

those results, they argued that Ta$_2$NiSe$_5$ would exhibit a blocking mechanism when pumped in the near-infrared regime, preventing a nonthermal structural phase transition. Apparently, our experimental observation of the PI-LR phase in Ta$_2$NiSe$_5$ is in sharp contrast to those results. The strong NIR laser pulses can definitely induce a structural phase transition. It is possible that the peak electric field of laser pulse used by Mor et al. was too small to drive a transition (see Supplementary Table 1). Another possibility is that the samples they used in their tr-ARPES study are thick bulk samples. As we mentioned above, the sharp switching is achieved currently only in thin flake samples. We note that more recent tr-ARPES measurements using an 800 nm amplified laser system also indicated a PI transient IMTs in Ta$_2$NiSe$_5$, though a stable state was not observed[24,25,49,50]. Those results are consistent with our observation of transient structural change at moderate excitations which can recovers to the initial state once the pump fluence is tuned back to a small value. Our present work brings new perspective to the control and manipulation of physical properties in Ta$_2$NiSe$_5$. The realization of ultrafast PI irreversible phase transition with a dramatic resistivity change will open up new prospects for designing novel functional photonic and electronic devices.

## Methods

**Sample growth**. The Ta$_2$NiSe$_5$ single crystals were synthesized by traditional chemical vapor transportation technology. The quartz tube loaded with the elements (99.99% Ta power, 99.99% Ni power, and 99.999% Se pellets) in a stoichiometric ratio together with additional iodine (2% of total mass) as the transport agent was vacuumed and flamed-sealed. Then the quartz tube was heated at 750–880 °C for 10 days in a two-zone furnace, followed by a natural cooling process. The obtained single crystals (normally in size of $5 \times 2 \times 0.1$ mm) were checked by the resistance, electron diffraction energy spectrum, and infrared spectroscopy.

**Device fabrication and temporal spectra of optical excitations**. The thin Ta$_2$NiSe$_5$ samples used in the experiments were mechanically exfoliated from the single crystals using the sticky tape and then transferred onto the cleaned sapphire substrates. The thickness of the films (10–50 nm in thickness) was measured by the KLA-Tencor's P-6 stylus profiler. The electrodes were fabricated by ultraviolet lithography using the bilayer photoresists (PMGI and AZ1500) and followed by the electron beam evaporation deposited with 10 nm Ti and 60 nm Au successively. The typical image of the devices taken from the Olympus BX51 Microscope.

The temporal resistance under photoexcitations were in situ recorded by the standard four-electrode method using a Keithley 2400 Source-Meter. A sample holder of the Liquid-He flow optical cryostat (Oxford Company) was used. A Ti:Sapphire Amplifier (800 nm, 35 fs, 1 kHz-repetition rate) was supplied for both the writing and pump–probe pulses. The shot-number of the writing pulse was controlled by a shutter. The temperature dependence of resistance $R(T)$ were also measured before and after the optical experiments by PPMS (Quantum Design Company). For optical pump–probe experiment, the two-electrodes geometry are normally used. The diameter of the writing-pulse spot was set normally in the range of 0.2–1 mm, fully covering the channel area between the electrodes. For pure optical pump–probe measurement, the diameter of the pump spot was 200–300 μm, more than two times larger than the one of probe pulse (90 μm). The PI phase transition can be achieved by both the front and backside incident.

The temporal reflectivity spectra $\Delta R/R(t)$ were measured by the 800–800 nm pump–probe geometry. The decay process was fitted by the two-exponential function $\Delta R/R(t) = A_1 \exp[-(t - t_0)/\tau_1] + A_2 \exp[-(t - t_0)/\tau_2] + y_0$, where $A_1$, $A_2$ were the amplitude of transient reflectance and $\tau_1$, $\tau_2$ were the relaxation time.

**TEM experiment**. A mechanical exfoliation method with scotch tape was used to obtain the Ta$_2$NiSe$_5$ nanoflakes, which were placed on 1000 mesh copper grids for TEM measurements. For the laser treatment, the nanoflakes were excited at temperature 50 K by a set of 30 pulses (800 nm, 35 fs, ~3.5 mJ/cm$^2$). In situ heating TEM observations were performed on a JEOL JEM-2100F microscope operated at 200 kV. The use of in situ heating TEM holder (model 652, Gatan Inc.) allows specimen temperature to be varied from room temperature to 750 K. Atomically-resolved HAADF images were obtained using a probe-corrected JEOL ARM-200C TEM. TEM bright-field imaging and selected area diffraction patterns were recorded using both ARM-200C and JEM-2100F microscopes.

## Data availability

The data that support the findings of this study are available from the corresponding author upon reasonable request.

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

## Acknowledgements

We would like to thank Profs. H. X. Yang and Z. Xu for their help in experiment and useful discussions. This work was supported by National Natural Science Foundation of China (Nos. 11888101, 11974019), the National Key Research and Development Program of China (Nos. 2017YFA0302904, 2017YFA0300303).

## Author contributions

D.W., Q.M.L., L.Y.S., and Z.X.W. performed crystal growth, device fabrication, and electrical property measurements. Q.M.L., D.W., L.Y.S., Z.X.W., S.J.Z., T.L., T.C.H., T.D., and N.L.W. performed optical experiments and analyzed the data. Z.A.L., H.F.T., and J.Q.L. performed TEM experiments and analyzed the data. D.W., Q.M.L., and N.L.W. wrote the paper, with contributions from all the authors. N.L.W. and D.W. conceived the project. All authors have approved the final version of the paper.

## Competing interests
The authors declare no competing interests.
