## [Peer Review File · Nature Communications]

REVIEWER COMMENTS

Reviewer #1 (Remarks to the Author):

Liu et al. present the generation of a quasi-permanent conducting state of Ta₂NiSe₅ (TNS) following single or multiple femtosecond laser pulses. This is an increasingly common phenomena that is only just beginning to be understood in the community. On first glance, this paper is very similar to reference 8, which is performed on 1T-TaS₂. Both a large and persistent change in resistivity occur and the phonon structure is modified. Visually Figure 1 of both papers is strikingly similar. However, unlike ref. 8, due to the stability of the photo-induced phase in TNS the authors can also obtain the structure of the photo-induced phase. This enables them to go beyond the speculation as to the origin of the effect. A clear modification to the stacking of the layers is observed and shows how the out-of-plane physics is highly important in these materials. These observations do not require more elaborate and exotic physics to explain and, in my opinion, raise questions on interpretations of other experiments that have not considered these effects in the dynamics such as 23 and 38.

However, I do find the paper could be structured better. I believe the paper will benefit from the structure imposed by Nature Communications and I would encourage the authors to work on this more and focus on their key observations.

Below I give my recommendations:

1. I found the discussion in the initial paragraph confusing and I think it would be beneficial if the author's state clearer that some phase transitions are transient, in the sense that they recover on the timescale of ms, and others are (quasi) permanent or meta-stable and require something other than time to reset them.
2. I would like to see a bit more analysis of the structure from the TEM images. In my opinion, it looks like Fig 3h is orientated 180 degrees relative to 3e (1-10 orientation) as the central 4 Ta atoms (the ones surrounded by Ni) point in the opposite direction. I think changing the orientation of the figure (mirroring in the y direction) would aid comparison, because at the moment it looks like the structure within the Ni layers is changing, when really it is the stacking of the Ta atoms between the Ni layers that is changing, or at least it looks that way to me. Comments on the nearest neighbor distances and angles of the Ta's within the Ni layers would make this clear. Furthermore, I do not see the word stacking fault in the text, yet it seems to be the most obvious word to describe why layers above and below the Ni layer have a different phase in the pristine and PI phases, is there a reason

why this term is not used? In addition, it would be good to know if the stacking faults are local, i.e. do some regions show the original stacking?

3. On this aspect, I think that the authors should put their work in context of ref 8 and other work on TiS_2 given the similarities. I want to draw the authors to Lee et al. Phys. Rev. Lett. 122, 106404 (2019), which is currently not cited, but I believe highly relevant to the work here. Lee describe the light induced phase transition in TaS_2 for Ref 8 in terms of stacking faults and the work presented in this paper seems to more-or-less confirm the scenario proposed by Lee, albeit for a different material. The stacking scenario seems more plausible than the “hidden quantum state” suggested in ref 8 and I believe that this work can comment on that.

4. I would also point out controlling the dynamics between layers of vdW materials is an active topic in other fields. For example, Ge diffusion into the vdW gap in Sb_2Te_3 -GeTe multi-layers is also known to dramatically change the electronic properties at room temperature and this can also be induced by a femtosecond light pulse at room temperature (see Kalikka et al. Nat. Commun. 7 11983 (2016) specifically and the literature on phase change materials). Discussing how the phase transition in TNS is, or is not, connected to phase change materials would also broaden its impact. In fact, here it is know that the both the magnitude of the laser-induced temperature jump and the cooling rate are vital for deciding which new phase is stably formed. A similarly strong dependence on the cooling rate is also predicted by Lee for TaS_2 . As attaching a thin sample to a substrate will dramatically change the cooling rate when compared to a single crystal, this may explain why only thin samples switch, rather than the peak electric field which is proposed by the authors. A comment on this would also be welcomed.

5. I also recommend the authors to expand more their comparison to other ultrafast work on this material. The authors state that thick samples do not show permanent switching, but perhaps they do transiently? Again in phase change materials, single crystals are also hard to switch to the amorphous phase, while it is easy in thin films. However, single crystals still transiently melt, but the rest of a crystal acts as seed to trigger re-crystallization, rather than amorphization which occurs in thin films. This mechanism is eluded to in the paper, but could some form of “melting” (not necessarily to a liquid phase) of the layer order perhaps generate the band dynamics seen in refs 23 and 38?

6. I found that the abstract did not really give the same impression as the paper. I did not see a multi-stage phase transition. It was not clear to me what each stage was? In fact, I did not see much evidence of a short-lived phase transition for weak intensities. Yes the phonon shows significant softening, but so does Bismuth. It wasn't clear to me that I should take this as a sign for a new phase. Clearly, the meta-stable state clearly is a different from the initial state, yet there is no mention of what I think is the key aspect of this work, that this is identified by a shifting of the Ta layers along the vertical direction. Thus I think this could be improved

One minor point Refs 3,4,9 and 13 are supposed to be related to meta-stable state in the manganites. Yet ref 4 is only a transient state. I would suggest Fiebig et al. Science 280, 1925 1998 is missing and could replace reference 4.

In conclusion, I think an adequately revised manuscript should be published in Nature Communications as I find the results very interesting, they go significantly beyond what has been measured to date, it is an emerging issue/opportunity in the field and has the potential to help unlock how and why these meta-stable phases form.

Reviewer #2 (Remarks to the Author):

Dear editor, dear authors,

I have worked on the present manuscript "Photoinduced multistage phase transitions in Ta₂NiSe₅" by QM Liu and coworker.

This is an interesting paper that reports on a photoinduced persistent phase transition in Ta₂NiSe₅. In my opinion the authors clearly show and convincingly proof the existence of such a transition. Besides the evidence of the transition bug transport measurements they also use a couple of techniques (coherent phonon spectroscopy and TEM) to characterize the light induced state and gain information on the nature of the phase transition.

A light induced persistent phase transition per se is not that spectacular anymore since meanwhile it is found in many different materials. So as long as there is no new mechanism that stabilizes the light induced phase I do not see the merit for a publication in Nature Communications. With by all means really beautiful data the authors do characterize the phonon driven lattice distortions as key element behind the light induced phase. As such I would tend to not recommend publication in Nature Comm. based on impact and novelty of the mechanism. The results itself are publishable for sure.

However the material under research here is Ta₂NiSe₅, a hot debated material that is believed to host an excitonic insulator state of condensed excitons, as the authors also remark. A lot of recent

literature (published and as preprints on arxiv) do discuss the possible driving mechanism behind a structural phase transition that appears together with the semiconductor-excitonic insulator transition in Ta₂NiSe₅. The data presented in the present manuscript in part can give important information to that puzzle and as such I believe the paper might have a crucial impact and therefore could be recommended for publication in Nat Comm.

In the end of their manuscript the authors already do make some links to the potential ExIns physics and also compare crucially to the semiconducting compound Ta₂NiS₅.

I would NOT recommend to rewrite the paper in the direction of solving the question of the ExIns problem and the nature of the phase transition (their data already shows that the effects they observe are not linked to the ExIns physics since both Ta₂NiSe₅ and Ta₂NiS₅ both show the same effects) but I would extend the discussion of the phonon spectroscopy and a bit the discussion in the conclusions on the importance of the lattice dynamics to the structural phase transition in Ta₂NiSe₅.

Here in particular comparison to new knowledge from Raman measurements that recently appeared as preprints might help and be very relevant in understanding the lattice and exciton dynamics under high laser fluences: MJ Kim et al. arxiv:2007.01723 and PA Volkov et al. arxiv:2007.07344. They discuss the low frequency Raman spectra in detail and in particular emphasize the role of the lattice and excitonic instabilities, respectively. A comparison and linked discussion of the results from Fig. 2 and Fig. 4 could help tremendously. In particular the role of the B_{2g} modes and the strong electronic backgrounds.

I believe that could push the paper over the top to gain impact and merit publication in Nature Comm. in my opinion.

Besides this more general view on the manuscript I do have a few more specific comments:

In the abstract the important TEM measurements are not mentioned.

On page 3 describing the electronic response the rise time says 100ps instead of I guess fs.

When describing these dynamics the authors may also want to frame the results to previous measurements like D Werdehausen et al J. Phys. Cond. Mat. 30, 305602 that have characterized the light induced melting dynamics before. Also there is a very nice new manuscript by P Andrich et al. arxiv2007.03368 that nicely extend such early measurements and also have an extended discussion on the phonon dynamics.

For the identification of the phonon modes I would also guide to the new Raman papers mentioned above (not only ref 25 claiming all A_g). In particular to the temperature driven phase transition the

nature of the modes above below the structural phase transition is important. As mentioned a more detailed description in comparison with the Raman works could be very beneficial.

In the discussion of the TEM figure: Which mode does the found displacement correspond to? Is it linked to some of the B2g type soft modes? Or not? Since it looks like the distortion is different from the thermal displacement it looks like that a new superlattice is induced, is that impression correct? And what would that mean?

As already mentioned for the discussion of the multipump studies in Fig 4 the link to the Raman behavior might be interesting. In particular with view on the induced broad backgrounds at high fluences could help to understand the influence of the lattice.

Since in equilibrium a lot of lattice dynamics seems to be clearly visible in particular at very high temperatures above T_c it would be interesting if these nice experiments could be extended beyond 350K base temperature.

Also could be sth said on a possible temperature dependence of the threshold here? Similar to the temperature dependent threshold that seems to link exciton physics with the lattice in ref. 21.

In the final discussion. As said here the comparison with the Exins physics is already made to some extent but maybe could be extended in analogy to the comparisons of Raman and coherent phonons suggested above.

I do have a problem in the authors claim that the exciton binding energy has to be small? If it is a charge transfer type of exciton like in many e.g. organic compounds then these binding energies can be high. However their wavefunctions do not overlap so therefore no condensation takes place. Here they seem to couple strongly to the lattice as literature shows. And disentangling these makes it so difficult to see if the lattice or the excitons do initially drive the condensation phase transition in the bulk.

Else I find it very important to show that Ta₂Ni₅S shows a similar PI phase transition so a direct link to the condensed exciton physics here can be ruled out.

Reviewer #3 (Remarks to the Author):

The draft "Photoinduced multistage phase transitions in Ta₂NiSe₅" present an experimental study of the photo-excited Ta₂NiSe₅ and shows the presence of a metastable phase beyond a certain pump threshold. I've read the paper with great interest as there are only a few examples of a long-lived metastable correlated material - notable examples are mentioned in the text examples, namely 1T-TaS₂ and manganite perovskites. While in last years, Ta₂NiSe₅ has been heavily debated due to the competition between the (possible) excitonic condensation and the strong electron-lattice coupling in the ground state, this paper opens a new direction for the photo-manipulation in this material.

My understanding of the main results. First, the observed metastable state induced by a short laser pulse is extremely stable up to high temperature and is distinguished from states in the equilibrium phase diagram. The phonon characterization with the pump-probe spectroscopy reveals a disappearance of the 2 THz phonon and a substantial modification of the 3.7 and 4 THz phonons. The TEM provides a crucial insight that the photo-excitation induces a large structural shift between the two nearest-neighbor chains. I find this valuable information.

The comparison with Ta₂NiS₅ serves as proof that the excitonic order does not play a major role in the metastability.

My main criticism of the paper is a lack of information and connection to previous literature:

- After the resistivity measurement, the focus of the study is on the properties of the lattice degrees of freedom. However, we do not learn anything about the electronic degrees of freedom. The resistivity drop for orders of magnitude is suggesting a dramatic redistribution of the electronic degrees of freedom. The authors should make a step further and provide some basic measurements. Optimally this would be either the optical response at the gap edge [as in PRB 95, 195144 (2017) for equilibrium] or PES data. But even more simple transport properties, like the Hall conductivity [see

SM of Ref. 23] or the thermopower [see Journal of the Physical Society of Japan 88, 113706 (2019)], could give valuable insight.

- There is quite some confusion in the study of this material as the same experimental probes gives quite contradicting results. For instance, t-ARPES is either showing a stable semiconducting phase [Ref. 23 and arXiv:2007.02909 - not cited] and a semimetallic state. At the end of your paper, you are suggesting that your measurement is more consistent with the later report, but I can not see any solid evidence for it as we do not know how electronic degrees of freedom are redistributed. This is important as the lattice is very indirectly excited via the electronic channel.

- You are suggesting that the photo-induced phase transition is a genuine collective response, which would imply that the energy/time scale for it is different than the energy/time scale for each element - in this case, phonons. This can be checked by modifying the pulse duration. In the text, you report about two numbers for the pump duration, namely 35 [main text] and 100 [caption Fig. 1] fs. Which one is correct? Later on, you say that the 100 fs pulse require a larger strength. How does the flip depend on the pump duration and can you flip the state by a much slower pump pulse that is comparable to the lattice oscillation, 250 fs for the 4 THz phonon or 0.5 ps for the 2 THz phonon ?

- comparison with the equilibrium structural transition. Several Raman studies have been performed on this material to characterize the equilibrium structural transition and there is no word about the comparison, see <https://arxiv.org/pdf/2007.08212.pdf>, <https://arxiv.org/pdf/2007.07344.pdf>]. These studies have identified the lattice excitations frequency with normal modes and it would be very useful for a reader to have this information in the text. As the most dramatic response in the equilibrium Raman measurement is exhibited by the 2 THz mode and it is also the main player in this study it would be important to know what are lattice motions corresponding to it. I'd strongly encourage authors to incorporate the comparison into the main text and compare with already available DFT studies, see Phys. Rev. Materials 4, 083601 (2020).

- I do not understand the statement "As the Ta lattice sliding has an intimate correlation with EI state in Ta₂NiSe₅, it can be expected that the pristine exciton condensation will be destroyed as soon as the Ta lattice transformation occurred. " The excitonic order formed between the Ta and Ni is linearly coupled to the in-chain shearing mode and this one is weakly modified [0.5 Å]. Are you referring to this distortion. If not, how could be the large distortion coupled to the excitonic order that is formed within the chain ? This is of course strongly correlated with my first question what happens to the electronic spectrum.

- this material is very anisotropic. Does the switching of the phase depend on the polarization of the pump pulse ? You only present the parallel to a-axis polarization.

- on a more speculative side: there is an interesting connection between your results with the hidden phase of 1T-TaS₂. A recent draft [Nature Communications volume 11, Article number: 1247 (2020)] is suggesting a photo-induced stacking in the non-conducting direction as the origin of this transition using the X-ray diffraction. Your result [shift] is essentially a 1D version of the 2D stacking in 1T-TaS₂. Can you comment on this analogy?

I believe that this draft does deserve a publication in Nat Comm, but only after a substantial modification of the draft as mentioned above.

typos:

- Fig2: pump-probe response

- Page 5: Considering the the large shear

We would like to express our great appreciation to all three referees for their careful reviews of our manuscript. We have studied referees' comments/suggestions carefully and have tried our best to improve our manuscript according to the comments. Below is a list of our detailed response to the reports by three referees.

Report of Reviewer #1

Liu et al. present the generation of a quasi-permanent conducting state of Ta₂NiSe₅ (TNS) following single or multiple femtosecond laser pulses. This is an increasingly common phenomena that is only just beginning to be understood in the community. On first glance, this paper is very similar to reference 8, which is performed on 1T-TaS₂. Both a large and persistent change in resistivity occur and the phonon structure is modified. Visually Figure 1 of both papers is strikingly similar. However, unlike ref. 8, due to the stability of the photo-induced phase in TNS the authors can also obtain the structure of the photo-induced phase. This enables them to go beyond the speculation as to the origin of the effect. A clear modification to the stacking of the layers is observed and shows how the out-of-plane physics is highly important in these materials. These observations do not require more elaborate and exotic physics to explain and, in my opinion, raise questions on interpretations of other experiments that have not considered these effects in the dynamics such as 23 and 38.

However, I do find the paper could be structured better. I believe the paper will benefit from the structure imposed by Nature Communications and I would encourage the authors to work on this more and focus on their key observations.

Below I give my recommendations:

1. I found the discussion in the initial paragraph confusing and I think it would be beneficial if the author's state clearer that some phase transitions are transient, in the sense that they recover on the timescale of ms, and others are (quasi) permanent or meta-stable and require something other than time to reset them.

-----response-----

We thank the referee for the comments. We modified the initial paragraph accordingly.

2. I would like to see a bit more analysis of the structure from the TEM images. In my opinion, it looks like Fig 3h is orientated 180 degrees relative to 3e (1-10 orientation) as the central 4 Ta atoms (the ones surrounded by Ni) point in the opposite direction. I think changing the orientation of the figure (mirroring in the y direction) would aid comparison, because at the moment it looks like the structure within the Ni layers is changing, when really it is the stacking of the Ta atoms between the Ni layers that is changing, or at least it looks that way to me. Comments on the nearest neighbor distances and angles of the Ta's

within the Ni layers would make this clear. Furthermore, I do not see the word stacking fault in the text, yet it seems to be the most obvious word to describe why layers above and below the Ni layer have a different phase in the pristine and PI phases, is there a reason why this term is not used? In addition, it would be good to know if the stacking faults are local, i.e. do some regions show the original stacking?

-----response-----

Here we illustrate the structural changes when mirroring the Fig. 3h by its y-direction, as seen in the attached figure below (the left is original and the right is the one of mirroring in the y direction). One can observe that the in-chain sliding of Ta atoms in the mirrored figure is still the prominent structural change compared with the original structure in Fig. 3e. In this Z-contrast imaging with atomic contrast of roughly Z^2 (Z refers to atomic number), the heavy Ta ($Z=73$) is mostly distinguishable compared with the Ni ($Z=28$) and Se ($Z=34$). Therefore, the Ta displacements after pulsed laser irradiation can be measured more precisely, while the dim contrasts of Ni and Se make the accurate determination of their changes difficult.

We mentioned that the main change from the TEM image appears to be a Ta lattice shear distortion along a-axis in the photoexcited LR phase. It is not really a stacking fault as observed in 1T-TaS₂ after photoexcitation, therefore we refrain from using the term “stacking fault”. From the TEM image along [110] direction, we estimate a counter Ta-Ta displacement roughly to be 0.5 ± 0.05 AA along $\langle 110 \rangle$ type direction. However, it is very difficult to measure the bond angle and distance change simply from a single projection image.

From the TEM morphology shown in Fig. 3c and 3f for the pristine and laser-pulse excited nano-flakes, we find rather homogeneous photoexcited region. The stripe-like domains disappear over a large area (the scale bar is 500 nm in Fig. 3c and 3f). The splitting and elongation of the diffraction peaks are absent. Nevertheless, we cannot rule out presence of local distortions.

3. On this aspect, I think that the authors should put their work in context of ref 8 and other work on TiS₂ given the similarities. I want to draw the authors to Lee et al. Phys. Rev. Lett. 122, 106404 (2019), which is currently not cited, but I believe highly relevant to the work here. Lee describe the light induced phase transition in TaS₂ for Ref 8 in terms of stacking faults and the work presented in this paper seems to more-or-less confirm the scenario proposed by Lee, albeit for a different material. The stacking scenario seems more plausible than the “hidden quantum state” suggested in ref 8 and I believe that this work can comment on that.

-----response-----

We cited the work by Lee et al. and added comments in the revised version. “Recent structure investigation combined with laser-pumping and first-principle calculation study on layered van der Waals CDW material 1T-TaS₂ suggested that the interlayer electronic ordering can play an important role in electronic phase transitions[34,35]. The metal-insulator transition is driven not by the 2D order itself, but by the vertical ordering or stacking of the 2D CDWs. Among 13 different stackings, there exist two exceptionally stable stacking configurations, one being an insulator and the other being a metal. They have a small energy difference, and their competition is responsible for the metal-insulator transition in 1T-TaS₂ [35]. The stacking scenario was used also to explain photoinduced insulator-metal phase transition in 1T-TaS₂ compound. However, the observed in-chain displacement of Ta-Ta atoms in Ta₂NiSe₅ in the photoinduced LR state indicated that the distortion appears more complicated than a simple stacking fault proposed in ref.[35]”.

4. I would also point out controlling the dynamics between layers of vdW materials is an active topic in other fields. For example, Ge diffusion into the vdW gap in Sb₂Te₃-GeTe multi-layers is also known to dramatically change the electronic properties at room temperature and this can also be induced by a femtosecond light pulse at room temperature (see Kalikka et al. Nat. Commun. 7 11983 (2016) specifically and the literature on phase change materials). Discussing how the phase transition in TNS is, or is not, connected to phase change materials would also broaden its impact. In fact, here it is know that the both the magnitude of the laser-induced temperature jump and the cooling rate are vital for deciding which new phase is stably formed. A similarly strong dependence on the cooling rate is also predicted by Lee for TaS₂. As attaching a thin sample to a substrate will dramatically change the cooling rate when compared to a single crystal, this may explain why only thin samples switch, rather than the peak electric field which is proposed by the authors. A comment on this would also be welcomed.

-----response-----

We cited the work by Kalikka et al. in the place when we address the issue that the stable phase transition was induced only in thin flake samples. “Switching of electronic properties more easily in thinner superstructure samples by femtosecond laser pulse has also been reported in other phase change materials, for example, in Ge₂Sb₂Te₅ [Kalikka] in terms of strain effect”. However, the pump fluence used by us is almost one order smaller than the

threshold by Kalikka. Simply, the temperature rise is not the reason for the observed phase transition because we observe one-direction irreversible LR phase transition. We did not have explicit experimental results showing that the laser heating and cooling rates are related to the observed structural phase transition. The difference between thin films and single crystals may be caused by the underlying stress. Therefore, we refrain from associating the observed phenomenon with the heating and cooling rates. In contrast, we find that the 100fs laser requires a larger fluence to induce the phase transition, so we think it is more likely related to the peak electric field.

5. I also recommend the authors to expand more their comparison to other ultrafast work on this material. The authors state that thick samples do not show permanent switching, but perhaps they do transiently? Again in phase change materials, single crystals are also hard to switch to the amorphous phase, while it is easy in thin films. However, single crystals still transiently melt, but the rest of a crystal acts as seed to trigger re-crystallization, rather than amorphization which occurs in thin films. This mechanism is eluded to in the paper, but could some form of “melting” (not necessarily to a liquid phase) of the layer order perhaps generate the band dynamics seen in refs 23 and 38?

-----response-----

We address this issue in the manuscript. We stated that “In Ta₂NiSe₅, the PI-LR state is always detected when the sample's thickness is thinner than the light penetration depth. However, for thicker bulk samples the 2.0- and 3.7-THz modes are usually recovered after the writing pulses ceased (see Fig. S6 in SM), with a high resistance state being kept as well. The observations are understandable. For the exfoliated thin flake sample transferred on the substrate with thickness smaller than the optical penetration depth, once the PI-LR state is triggered, the underlying lattice would be completely transformed into a new phase that is weakly affected from the substrate. Whereas for a thick bulk sample, the underneath unexcited or incompletely transitioned lattice may inevitably exert a restoring strain to the upper excited layers, resulting in a possible recovery of the 2.0- and 3.7-THz modes when the pump fluence is below the damage threshold.”

6. I found that the abstract did not really give the same impression as the paper. I did not see a multi-stage phase transition. It was not clear to me what each stage was? In fact, I did not see much evidence of a short-lived phase transition for weak intensities. Yes the phonon shows significant softening, but so does Bismuth. It wasn't clear to me that I should take this as a sign for a new phase. Clearly, the meta-stable state clearly is a different from the initial state, yet there is no mention of what I think is the key aspect of this work, that this is identified by a shifting of the Ta layers along the vertical direction. Thus I think this could be improved.

-----response-----

By multistage, we mean that “Upon excitation by weak pulse intensity, the system is triggered to a short-lived state accompanied by a structural change. Further increasing the excitation intensity beyond a threshold, a photoinduced steady new state is achieved

where the resistivity drops by more than four orders at temperature 50 K”.

In the main text, we have following description: “Initially at weak pump fluence, the primitive five coherent modes are recorded. With the pump fluence increasing, the 2.0-THz mode tends degraded gradually and disappear completely above $\sim 550 \mu\text{J}/\text{cm}^2$, while the 3.7- and 4.0 THz modes get merged into a broad one, suggesting a PI structural phase transition. However this transition is transient, it doesn't yield a permanent reduction of DC resistivity and the coherent phonon spectra will recover to the initial state when the pump fluence is tuned back to small values (See Fig. S8 in SM). ...”. Our main interest and focus in this work is the non-reversible phase transition at high fluence. We showed that there is a threshold of about $2.5 \text{ mJ}/\text{cm}^2$ for driving this steady new state.

One minor point Refs 3,4,9 and 13 are supposed to be related to meta-stable state in the manganites. Yet ref 4 is only a transient state. I would suggest Fiebig et al. Science 280, 1925 1998 is missing and could replace reference 4.

-----response-----

We made change accordingly.

In conclusion, I think an adequately revised manuscript should be published in Nature Communications as I find the results very interesting, they go significantly beyond what has been measured to date, it is an emerging issue/opportunity in the field and has the potential to help unlock how and why these meta-stable phases form.

-----response-----

We thank the referee for the kind recommendation.

Reviewer #2 (Remarks to the Author):

Dear editor, dear authors,

I have worked on the present manuscript “Photoinduced multistage phase transitions in Ta₂NiSe₅” by QM Liu and coworker.

This is an interesting paper that reports on a photoinduced persistent phase transition in Ta₂NiSe₅. In my opinion the authors clearly show and convincingly proof the existence of such a transition. Besides the evidence of the transition bug transport measurements they also use a couple of techniques (coherent phonon spectroscopy and TEM) to characterize the light induced state and gain information on the nature of the phase transition.

A light induced persistent phase transition per se is not that spectacular anymore since

meanwhile it is found in many different materials. So as long as there is no new mechanism that stabilizes the light induced phase I do not see the merit for a publication in Nature Communications. With by all means really beautiful data the authors do characterize the phonon driven lattice distortions as key element behind the light induced phase. As such I would tend to not recommend publication in Nature Comm. based on impact and novelty of the mechanism. The results itself are publishable for sure.

However the material under research here is Ta₂NiSe₅, a hot debated material that is believed to host an excitonic insulator state of condensed excitons, as the authors also remark. A lot of recent literature (published and as preprints on arxiv) do discuss the possible driving mechanism behind a structural phase transition that appears together with the semiconductor-excitonic insulator transition in Ta₂NiSe₅. The data presented in the present manuscript in part can give important information to that puzzle and as such I believe the paper might have a crucial impact and therefore could be recommended for publication in Nat Comm.

In the end of their manuscript the authors already do make some links to the potential ExIns physics and also compare crucially to the semiconducting compound Ta₂NiS₅.

I would NOT recommend to rewrite the paper in the direction of solving the question of the ExIns problem and the nature of the phase transition (their data already shows that the effects they observe are not linked to the ExIns physics since both Ta₂NiSe₅ and Ta₂NiS₅ both show the same effects) but I would extend the discussion of the phonon spectroscopy and a bit the discussion in the conclusions on the importance of the lattice dynamics to the structural phase transition in Ta₂NiSe₅.

Here in particular comparison to new knowledge from Raman measurements that recently appeared as preprints might help and be very relevant in understanding the lattice and exciton dynamics under high laser fluences: MJ Kim et al. arxiv:2007.01723 and PA Volkov et al. arxiv:2007.07344. They discuss the low frequency Raman spectra in detail and in particular emphasize the role of the lattice and excitonic instabilities, respectively. A comparison and linked discussion of the results from Fig. 2 and Fig. 4 could help tremendously. In particular the role of the B_{2g} modes and the strong electronic backgrounds.

-----response-----

We thank the referee for the comments. We cited the work by MJ Kim et al. and PA Volkov et al. in the revised version. According to these reports, the B_{2g} modes in high-temperature orthorhombic phase would evolve into A_g symmetry in low-temperature monoclinic phase. A recent theoretical work (A. Subedi. Phys. Rev. Materials 4, 083601 (2020)) has indicated the significant role of the B_{2g} mode to this phase transition. The low temperature 2-THz mode is in A_g channel (while above T_c, it is in B_{2g} symmetry) which corresponds to the in-chain Ta atoms oscillations. By Raman measurement, a strong T-dependent electronic background broadening near the 2-THz B_{2g} mode above T_c have been commonly detected by different groups. When T_c is approached from high temperature, a softening of 2-THz B_{2g} mode was observed by MJ Kim et al. but not others. Those results led to conclusions of either a purely

electronic or a cooperative structural and electronic origin about the phase transition.

Here in our study, our optical pump-probe spectra detect only the Ag modes. So in the pristine state below T_c , five Ag modes were detected at 50K and two of them (2.0- and 4.0-THz) got disappeared due to evolving into B2g modes above T_c , the result is in accordance with the Raman spectra in the (X-) geometry below T_c and in the (XX) geometry above T_c . Yet with excitation fluence increasing, we observed that the 2-THz mode get broaden till disappear. The disappearance of 2-THz under high excitation fluence suggests a photo-induced phase transition. Our TEM measurements further confirmed the in-chain Ta atoms displacement. The result corresponds well with the scenario of the 2-THz issue proposed by the Raman references. However, the feature of large displacement of Ta atoms by our TEM provided evidence that the PI new state is different from any of already known phases in pristine Ta₂NiSe₅.

In the revised version and following the TEM discussion, we have added the following new paragraph/description: “It would be interesting to compare the above coherent phonon excitations and structural characterization with the equilibrium structural transition revealed by Raman scattering studies [29–32]. Raman studies have revealed significant role played by the 2 THz mode in the structural phase transition. The mode has a B2g symmetry in the high temperature orthorhombic phase but evolves into an Ag symmetry in the low-temperature monoclinic phase. It corresponds to the in-chain Ta lattice oscillations or shear motion of TaSe₆ octahedra along the a-axis[Subedi]. An instability of this mode as the origin of phase transition was suggested by density function theory based calculations and indeed observed in a recent Raman measurement. In the present study, the optical pump-probe experiment detects only the Ag coherent phonon modes. In the pristine state below T_c , five Ag modes were detected at 50 K and two of them (2.0- and 4.0- THz) got disappeared due to evolving into B2g modes above T_c , the result is in accordance with the Raman measurement. On the other hand, the 2-THz mode is also absent in the PI-LR phase. In fact, the pump fluence dependent pump-probe measurement in the low-temperature monoclinic phase presented below indicates a clear weakening and disappearance of the 2-THz mode upon increasing fluence. The in-chain Ta atoms displacement found by TEM agrees well with the lattice instability of B2g mode proposed by the Raman work. Nevertheless, the large displacement of Ta atoms revealed by TEM measurement yields evidence that the PI new state is different from any of already known phases in pristine Ta₂NiSe₅.”

I believe that could push the paper over the top to gain impact and merit publication in Nature Comm. in my opinion.

Besides this more general view on the manuscript I do have a few more specific comments:

In the abstract the important TEM measurements are not mentioned.

-----response-----

We have modified the abstract.

On page 3 describing the electronic response the rise time says 100ps instead of I guess fs.
-----response-----

We corrected the typo. Yes, it is fs.

When describing these dynamics the authors may also want to frame the results to previous measurements like D Werdehausen et al J. Phys. Cond. Mat. 30, 305602 that have characterized the light induced melting dynamics before. Also there is a very nice new manuscript by P Andrich et al. arxiv2007.03368 that nicely extend such early measurements and also have an extended discussion on the phonon dynamics.

-----response-----

We cited these two works and modified the description in the revised version as: “Finally, our work reveals a significantly different photoinduced effect from reported work for the compound. The issue of photoinduced phase transition in Ta₂NiSe₅ has been heavily debated with controversial results. Several reports denied a possible of photoinduced phase transition [23, 44, 45], however some recent reports support that transition [24, 25, 46, 47]. For example, in earlier tr-ARPES and ultrafast pump-probe studies on Ta₂NiSe₅[23, 44], Mor et al. reported that the photon absorption saturated at a critical fluence of $F_c=0.2$ mJ/cm² upon photoexcitation, and above the saturation fluence the band energy gaps widened, reflecting an increase of the exciton condensate density in the EI phase in Ta₂NiSe₅[23]. In accord with the tr-ARPES results, their pump-probe measurement also indicated a saturation phenomenon above a critical excitation fluence [44]. Based on those results, they argued that Ta₂NiSe₅ would exhibit a blocking mechanism when pumped in the near-infrared regime, preventing a nonthermal structural phase transition. Apparently, our experimental observation of the PI-LR phase in Ta₂NiSe₅ is in sharp contrast to those results. The strong NIR laser pulses can definitely induce a structural phase transition. It is possible that the peak electric field of laser pulse used by Mor et al. was too small to drive a transition. Another possibility is that the samples they used in their tr-ARPES study are thick bulk samples. As we mentioned above, the sharp switching is achieved currently only in thin flake samples. We note that more recent tr-ARPES measurements using an 800 nm amplified laser system also indicated a photo-induced transient insulator-to-metal transitions in Ta₂NiSe₅, though a stable state was not observed [24, 25, 46, 47]. Those results are consistent with our observation of transient structural change at moderate excitations which can recovers to the initial state once the pump fluence is tuned back to a small value. Our present work brings new perspective to the control and manipulation of physical properties in Ta₂NiSe₅. The realization of ultrafast photoinduced irreversible phase transition with a dramatic resistivity change will open up new prospects for designing novel functional photonic and electronic devices.”

For an easy browse, a list of pump laser parameters from different references is provided:

Group	Centre Wavelength	Pulse Duration	Repetition Frequency	Max. Pump Fluence (mJ/cm ²)	Electric Field Peak (MV/cm)	PI-phase Transition	Experiment type
Ref.23	800 nm	110fs	40kHz	0.47	1.8	No	t-ARPES
arXiv:2007.02909	800 nm	230 fs	100 kHz	0.85	1.67	No	t-ARPES
Ref.45	800 nm	130 fs	/	1.75	3.18	No	fs optical spectra
Ref. 24	800nm	30 fs	1 kHz	1.56	6.26	Yes	t-ARPES
Ref.25	800 nm	35 fs	/	2.27	7.02	Yes	t-ARPES
Ref.46	700 nm	30 fs	500 kHz	0.32	2.83	Yes	t-ARPES
Ref.47	700 nm	12 fs	/	0.88	7.43	Yes	fs optical spectra
The present work	800 nm	35 fs	1 kHz	3.5	8.68	Yes	fs optical spectra

For the identification of the phonon modes I would also guide to the new Raman papers mentioned above (not only ref 25 claiming all Ag). In particular to the temperature driven phase transition the nature of the modes above below the structural phase transition is important. As mentioned a more detailed description in comparison with the Raman works could be very beneficial.

-----response-----

We cited more Raman works and added comments in the revised version, as already mentioned in above response.

In the discussion of the TEM figure: Which mode does the found displacement correspond to? Is it linked to some of the B2g type soft modes? Or not? Since it looks like the distortion is different from the thermal displacement it looks like that a new superlattice is induced, is that impression correct? And what would that mean?

-----response-----

The 2-THz mode, has Ag symmetry below Tc and B2g symmetry above Tc, corresponds to the Ta atoms in-chain oscillations. The in-chain Ta atoms displacement found by TEM agrees well with the lattice instability of B2g mode proposed by the Raman work. Nevertheless, the large displacement of Ta atoms revealed by TEM indicates that the new phase is different from any of known phases available via thermal process.

As already mentioned for the discussion of the multipump studies in Fig 4 the link to the Raman behavior might be interesting. In particular with view on the induced broad backgrounds at high fluences could help to understand the influence of the lattice.

-----response-----

The indication of Raman behavior to our observations has discussed in previous comments.

Here, for the Fig 4 with broad backgrounds, we have the description: “Fig. 4b presents the coherent phonon spectra in a pristine nano-thickness sample with an increase of pump fluence up to 2.2 mJ/cm². The original oscillations in the time domain is presented in Fig. S7 in SM. Initially at weak pump fluence, the primitive five coherent modes are recorded. With the pump fluence increasing, the 2.0-THz mode tends degraded gradually and disappear completely above $\sim 550 \mu\text{J}/\text{cm}^2$, while the 3.7- and 4.0 THz modes get merged into a broad one, suggesting a PI structural phase transition. However, this transition is transient, it doesn't yield a permanent reduction of DC resistivity and the coherent phonon spectra will recover to the initial state when the pump fluence is tuned back to small values (See Fig. S8 in SM). With the excitation intensity increasing, the three low frequency modes show a clear red shift and an asymmetric broadening to the lower-frequency side, the phenomena can be ascribed to coherent anharmonic oscillations under high density photoexcitations [37, 38]. The red shift of the peak frequency can be considered as an electronic softening effect relating with the transient photoexcited carriers. While for the asymmetric broadening modes, the effect is typically referred as the nonthermal modification to the lattice potential-energy surface at high intensity excitations [39, 40]. This broadening modification is a transient effect that only lasts for a short time before thermal equilibrium is attained between electrons and lattice background. We performed the time-domain trace analysis for the modification effect. Fig. 4c shows the Fourier transformed spectra obtained after cutting the initial time delay T_d at two different positions for a pump fluence 1.5 mJ/cm². As can be seen, if we perform Fourier transformation of the pump-probe waveform after the first 2 ps ($T_d = 2$ ps), the red shift and broadening of the phonon modes approach to disappear. The inset shows the change of 3 THz mode of Fourier transformed spectra after cutting different time delays T_d . A more detailed analysis of the transient phonon spectra as a function of cutting time delay T_d at various excitation intensity is presented in Fig. S9 in SM.”

The case of broadening modification in Raman spectrum was found to occur suddenly as soon as the temperature rises across T_c (and only broadened in high-temperature phase). It is an adiabatic process and occurs in the thermal equilibrium states. However, it is a transient and non-equilibrium process in Fig.4 of our work. Typically to Fig 4, it is hard for us to link it with the mentioned Raman observations.

Since in equilibrium a lot of lattice dynamics seems to be clearly visible in particular at very high temperatures above T_c it would be interesting if these nice experiments could be extended beyond 350K base temperature.

-----response-----

For the optical pump-probe spectroscopy, it in principle can only detect Ag modes (and to Ta₂NiSe₅, it needs to operate at low temperature below ~ 250 K to detect and track 2-THz mode). Therefore, it is not expected to detect signal of the 2-THz mode. Additionally, our cryostat cannot go beyond 350 K at the moment.

Also could be sth said on a possible temperature dependence of the threshold here? Similar

to the temperature dependent threshold that seems to link exciton physics with the lattice in ref. 21.

-----response-----

For the issue of temperature dependent threshold, we regret that we haven't find an effective way to provide a systematic study on it. To be precision, it is better to operate the threshold estimation on one same sample. However, the photo-induced LR state is non-reversible, that makes a sample only single-used once the threshold is achieved.

In the final discussion. As said here the comparison with the Exins physics is already made to some extend but maybe could be extended in analogy to the comparisons of Raman and coherent phonons suggested above.

-----response-----

As mentioned in the above response, we have tried to provide comparisons to the Raman measurement to the best of our knowledge to Ta₂NiSe₅ system.

I do have a problem in the authors claim that the exciton binding energy has to be small? If it is a charge transfer type of exciton like in may e.g. organic compounds then these binding energies can be high. However their wavefuctions do not overlap so therefore no condensation takes place. Here they seem to couple strongly to the lattice as literature shows. And disentangling these makes it so difficult to see if the lattice or the excitons do initially drive the condensation phase transition in the bulk.

-----response-----

We mentioned that "the binding energy of exciton appears to be very sensitive to the lattice distortion". As is known, for a small gap semiconductor, an excitonic instability occurs if the exciton binding energy is larger than the electronic band energy gap. Then, the system would form a new phase in which the new exciton binding energy become smaller than the new band energy gap till the system get stabilized. For the photoinduced LR phase, "the exciton binding energy in this phase must be very small because of the absence of phase transition." That is what we mean when discussing the exciton binging energy.

Else I find it very important to show that Ta₂NiS₅ shows a similar PI phase transition so a direct link to the condensed exciton physics here can be ruled out.

-----response-----

We thank again the referee for the comments. Yes, the work on Ta₂NiS₅ indicates that the exciton interaction is not a prerequisite for the formation of PI-LR state.

Reviewer #3 (Remarks to the Author):

The draft "Photoinduced multistage phase transitions in Ta₂NiSe₅" present an experimental study of the photo-excited Ta₂NiSe₅ and shows the presence of a metastable phase beyond a certain pump threshold. I've read the paper with great interest as there are only a few examples of a long-lived metastable correlated material - notable examples are mentioned in the text examples, namely 1T-TaS₂ and manganite perovskites. While in last years, Ta₂NiSe₅ has been heavily debated due to the competition between the (possible) excitonic condensation and the strong electron-lattice coupling in the ground state, this paper opens a new direction for the photo-manipulation in this material.

My understanding of the main results. First, the observed metastable state induced by a short laser pulse is extremely stable up to high temperature and is distinguished from states in the equilibrium phase diagram. The phonon characterization with the pump-probe spectroscopy reveals a disappearance of the 2 THz phonon and a substantial modification of the 3.7 and 4 THz phonons. The TEM provides a crucial insight that the photo-excitation induces a large structural shift between the two nearest-neighbor chains. I find this valuable information.

The comparison with Ta₂NiS₅ serves as proof that the excitonic order does not play a major role in the metastability.

My main criticism of the paper is a lack of information and connection to previous literature:

- After the resistivity measurement, the focus of the study is on the properties of the lattice degrees of freedom. However, we do not learn anything about the electronic degrees of freedom. The resistivity drop for orders of magnitude is suggesting a dramatic redistribution of the electronic degrees of freedom. The authors should make a step further and provide some basic measurements. Optimally this would be either the optical response at the gap edge [as in PRB 95, 195144 (2017) for equilibrium] or PES data. But even more simple transport properties, like the Hall conductivity [see SM of Ref. 23] or the thermopower [see Journal of the Physical Society of Japan 88, 113706 (2019)], could give valuable insight.

-----response-----

We thank the referee for the comments. Actually, the experiments of using optical response, PES or other transport studies to characterize the photo-induced effects on nano-flake Ta₂NiSe₅ samples would be extremely challenging to us. We tend to leave these subjects to experts with special expertise on those techniques on nano-flake samples in the field, or to see possibility to collaborate with those experts in the future.

- There is quite some confusion in the study of this material as the same experimental probes gives quite contradicting results. For instance, t-ARPES is either showing a stable semiconducting phase [Ref. 23 and arXiv:2007.02909 - not cited] or a semimetallic state. At

the end of your paper, you are suggesting that your measurement is more consistent with the later report, but I can't see any solid evidence for it as we do not know how electronic degrees of freedom are redistributed. This is important as the lattice is very indirectly excited via the electronic channel.

-----response-----

We thank the referee for the comments. The mentioned discrepancies might come from the different laser pulse conditions. It is possible that the peak electric field of laser pulse used by [Ref. 23 and arXiv:2007.02909] was too small to drive a transition. Below is the list of pump laser parameters from different references.

Comparatively, the pump lasers used by our present work are more similar to the ones used by Ref.24, 25, where the photoinduced semimetallic state were detected. Also, by analysis of the temporal pump-probe response spectra, the dynamics of the electronic part (the fast decay process in Fig.2a and corresponding discussion) is in accordance well with the electronic pre-thermalization process revealed by t-ARPES experiments [Ref.24, 25,40]. Thus, though our measurements had no direct evidence for the electronic degrees of freedom, we suggest that our measurement is more consistent with the later reports.

Group	Centre Wavelength	Pulse Duration	Repetition Frequency	Max. Pump Fluence (mJ/cm ²)	Electric Field Peak (MV/cm)	PI-phase Transition	Experiment type
Ref.23	800 nm	110fs	40kHz	0.47	1.8	No	t-ARPES
arXiv:2007.02909	800 nm	230 fs	100 kHz	0.85	1.67	No	t-ARPES
Ref.45	800 nm	130 fs	/	1.75	3.18	No	fs optical spectra
Ref. 24	800nm	30 fs	1 kHz	1.56	6.26	Yes	t-ARPES
Ref.25	800 nm	35 fs	/	2.27	7.02	Yes	t-ARPES
Ref.46	700 nm	30 fs	500 kHz	0.32	2.83	Yes	t-ARPES
Ref.47	700 nm	12 fs	/	0.88	7.43	Yes	fs optical spectra
The present work	800 nm	35 fs	1 kHz	3.5	8.68	Yes	fs optical spectra

- You are suggesting that the photo-induced phase transition is a genuine collective response, which would imply that the energy/time scale for it is different than the energy/time scale for each element - in this case, phonons. This can be checked by modifying the pulse duration. In the text, you report about two numbers for the pump duration, namely 35 [main text] and 100 [caption Fig. 1] fs. Which one is correct? Later on, you say that the 100fs pulse require a larger strength. How does the flip depend on the pump duration and can you flip the state by a much slower pump pulse that is comparable to the lattice oscillation, 250 fs for the 4 THz phonon or 0.5ps for the 2 THz phonon?

-----response-----

We modified that typo of the fluence, in the caption Fig.1, it should be 35 fs. For the issue of

long pulse duration, though we noticed that the PI-LR phase can be driven by laser pulses of 100 fs, we didn't investigate the pulse duration dependence systematically and quantitatively (That will cost lots of samples because of the irreversible phase transition, and more indeterminacies might arise for different samples used).

For the issue of "the slower pump pulses comparable to the lattice oscillation". In principle, it depends mainly on the probe (but not the pump) pulse duration in differentiating the phonon oscillations, i.e. probe pulse will be required roughly < 250 fs for 4 THz; < 0.5 ps for 2 THz. But we haven't tried that slow pulses (250 fs; 0.5 ps) in our experiments.

- comparison with the equilibrium structural transition. Several Raman studies have been performed on this material to characterize the equilibrium structural transition and there is no word about the comparison, see <https://arxiv.org/pdf/2007.08212.pdf>, <https://arxiv.org/pdf/2007.07344.pdf>. These studies have identified the lattice excitations frequency with normal modes and it would be very useful for a reader to have this information in the text. As the most dramatic response in the equilibrium Raman measurement is exhibited by the 2 THz mode and it is also the main player in this study it would be important to know what are lattice motions corresponding to it. I'd strongly encourage authors to incorporate the comparison into the main text and compare with already available DFT studies, see Phys. Rev. Materials 4, 083601 (2020).

-----response-----

As previous (the first) response to the Referee#2, we have provided such the Raman comparisons to the best of our knowledge to Ta₂NiSe₅ system. We cited above works and added the following new paragraph/description: "It would be interesting to compare the above coherent phonon excitations and structural characterization with the equilibrium structural transition revealed by Raman scattering studies [29–32]. Raman studies have revealed significant role played by the 2 THz mode in the structural phase transition. The mode has a B_{2g} symmetry in the high temperature orthorhombic phase but evolves into an A_g symmetry in the low-temperature monoclinic phase. It corresponds to the in-chain Ta lattice oscillations or shear motion of TaSe₆ octahedra along the a-axis[Subedi]. An instability of this mode as the origin of phase transition was suggested by density function theory based calculations and indeed observed in a recent Raman measurement. In the present study, the optical pump-probe experiment detects only the A_g coherent phonon modes. In the pristine state below T_c, five A_g modes were detected at 50 K and two of them (2.0- and 4.0- THz) got disappeared due to evolving into B_{2g} modes above T_c, the result is in accordance with the Raman measurement. On the other hand, the 2-THz mode is also absent in the PI-LR phase. In fact, the pump fluence dependent pump-probe measurement in the low-temperature monoclinic phase presented below indicates a clear weakening and disappearance of the 2-THz mode upon increasing fluence. The in-chain Ta atoms displacement found by TEM agrees well with the lattice instability of B_{2g} mode proposed by the Raman work. Nevertheless, the large displacement of Ta atoms revealed by TEM measurement yields evidence that the PI new state is different from any of already known phases in pristine Ta₂NiSe₅"

- I do not understand the statement "As the Ta lattice sliding has an intimate correlation with EI state in Ta₂NiSe₅, it can be expected that the pristine exciton condensation will be destroyed as soon as the Ta lattice transformation occurred. " The excitonic order formed between the Ta and Ni is linearly coupled to the in-chain shearing mode and this one is weakly modified [0.5 Å]. Are you referring to this distortion ? If not, how could be the large distortion coupled to the excitonic order that is formed within the chain? This is of course strongly correlated with my first question what happens to the electronic spectrum.

-----response-----

The scale [0.5 Å] is actually not a small number for Ta-lattice distortion but a large number. Comparatively, the 326 K phase transition in pristine Ta₂NiSe₅ only corresponds with a very tiny Ta-lattice modification far less than [0.1 Å].

- this material is very anisotropic. Does the switching of the phase depend on the polarization of the pump pulse? You only present the parallel to a-axis polarization.

-----response-----

Yes, we also found that the LR state can be induced by other polarizations of the pump pulse.

- on a more speculative side: there is an interesting connection between your results with the hidden phase of 1T-TaS₂. A recent draft [Nature Communications volume 11, Article number: 1247 (2020)] is suggesting a photo-induced stacking in the non-conducting direction as the origin of this transition using the X-ray diffraction. Your result [shift] is essentially a 1D version of the 2D stacking in 1T-TaS₂. Can you comment on this analogy?

-----response-----

(As the response for this stacking issue to the first Referee) We cited this work and added comments in the revised version. "Recent structure investigation combined with laser-pumping and first-principle calculation study on layered van der Waals CDW material 1T-TaS₂ suggested that the interlayer electronic ordering can play an important role in electronic phase transitions[34,35]. The metal-insulator transition is driven not by the 2D order itself, but by the vertical ordering or stacking of the 2D CDWs. Among 13 different stackings, there exist two exceptionally stable stacking configurations, one being an insulator and the other being a metal. They have a small energy difference, and their competition is responsible for the metal-insulator transition in 1T-TaS₂ [35]. The stacking scenario was used also to explain photoinduced insulator-metal phase transition in 1T-TaS₂ compound. However, the observed in-chain displacement of Ta-Ta atoms in Ta₂NiSe₅ in the photoinduced LR state indicated that the distortion appears more complicated than a simple stacking fault proposed in ref.35"

I believe that this draft does deserve a publication in Nat Comm, but only after a substantial modification of the draft as mentioned above.

typos:

- Fig2: pump-probe response
- Page 5: Considering the the large shear

-----response-----

We thank the referee for the comment. We went through the manuscript and made corrections.

List of major changes:

1. We added a new paragraph addressing/comparing the stacking scenario (proposed to explain the laser-induced phase transition in 1T-TaS₂).
2. We added a new paragraph to make comparison with reported Raman measurement.
3. Minor changes/typo corrections were made in other places in the manuscript. (those changes in the manuscript were shown as blue colour).
4. All the references mentioned by the referees were added.

REVIEWERS' COMMENTS

Reviewer #1 (Remarks to the Author):

The revised manuscript by Liu et al is an improvement on the initial submission. I still feel that it could be improved. In particular, although comments on the literature I suggested have been added, it feels it has been done more to satisfy me, rather than to improve the paper. i.e. the papers have not had an real influence on the general discussion in the paper. If the authors do not think these papers add anything interesting to the paper they do not need to be added. I agree with the authors that this appears to be more complicated than simple stacking faults, but is that complication really important? Perhaps that is the case in TaS₂ too, it has not been explicitly measured. Such things could be commented on.

Ultimately, I am a little unsure of if the authors are trying to communicate about this phase transition. On the one hand, the results are clear and almost speak for themselves, especially with regards to the excitonic insulator. In particular, I do not know if the authors are claiming this process is driven by a particular phonon or peak field, or something else.

That said, although I would encourage the authors to try to focus the argument a little more, I would also be happy to accept it in the current form. However, in any case, the paper needs to be checked as I found several mistakes of which only a few are highlighted here:

Ultrashort laser pulse not only is a powerful -> Ultrashort laser pulses are not only a powerful...

The samples, in the thickness of -> The samples, with thicknesses of

Two of them (2.0- and 4.0- THz) got disappeared -> two of them (2.0- and 4.0- THz) disappeared

Is EI defined? It is mentioned twice but I cannot see a definition

Reviewer #2 (Remarks to the Author):

Dear editor, dear authors,

thanks for getting back to me with the revised manuscript. As said in the previous round of reports the finding of the phase switching effect alone I do not judge as exciting enough to recommend publication in Nat. Comm.

Also in the new manuscript there is no additional information on the nature of the phase transition that would prove a novel type of transition.

However, as mentioned I do find the material system Ta₂NiSe₅ highly interesting and the nature of structural phase transition in that material is under hot discussions both both from experimental and theoretical points of view.

In the new manuscript the authors link to the suggested additional pre-prints and papers I did suggest and discuss their own results in this context. As such I feel that the paper adds important information to a question of immediate interest in the community on a very decisive question on the phase transition in Ta₂NiSe₅.

Therefore I do recommend publication of the present manuscript, now.

I also did read through the responses to the other reports and find the answers of the author sufficient. Also the answers the open points and "rejections" to requests on additional data that would be nice to have but that go clearly beyond the scope of the present study or are not easily gained experimentally is done well and justified. The latter also counts for requests made by myself.

Some minor points that may or may not be helpful:

(1) to the report of reviewer 1:

In the question of presenting figure 3e/h. I would not mirror the image. The red line in the data made it clear how to read the figure - at least for me. Maybe as a compromise one could align the red line in both figures to the same position and show a slightly larger cut in the figures.

(2) To my point on the discussion of the TEM figure. I am fine with the response they give to me in the letter. Maybe the authors could also add this statement to the main text.

In their revised manuscript they just write "the above coherent phonon excitation"....

In detail there is an offset displacement that is in the direction of the phonon mode, the oscillatory motion would be then on top of that.

(3) On the point of the discussion of the exciton binding energy. I do understand the points of the authors. However that is what one would theoretically expect as a scenario. But there is no experimental evidence for the exciton binding energy/exciton radius.

Maybe instead of "apparently, the ex binding energy must be...." the others could tone down a bit to "The absence of a phase transition suggests or is compatible with...."

Reviewer #3 (Remarks to the Author):

Authors have made connections to Raman, and the stacking in 1T-TaS₂ problem and appropriately modify/improve the main text.

I can imagine that the measurement of less trivial transport properties, like the Hall conductivity or the thermopower, is challenging for authors. However, I would expect a more proactive approach to analyzing the pulse duration as it would provide additional insights into the electronic-lattice interplay. The lack of the latter is still my main criticism of the paper.

I like the summary table on different transient experiments that authors have provided in the reply. I believe it would be useful to include it in the SM and point a reader to it due to the "confusion in the field".

A side remark. I've rechecked reference 23 [Mor et al.], and your classification could be a bit misleading [or the text in their paper]. Indeed, authors measure the saturation effect, but on Fig.3 we see that on the timescale longer than ~ 1 ps the peak position shrinks below the equilibrium value (at least a partial gap closure) and the EDC data in the SM [Fig S2] shows a strong signal at the chemical potential. Unfortunately, authors do not show high-excitation and long time data, but I strongly suspect that they look semimetallic-like. I think the discussion in conclusion ["above the saturation fluence the band energy gaps widened, reflecting an increase of the exciton condensate density in the EI phase in Ta₂NiSe₅ [23] etc."] is still fine, but you could comment on the issue.

I'd also suggest that authors mention in the main text that any polarization can induce PI state as this is a nontrivial effect.

If I sum up, I would recommend the publication as the connection with the Raman and the stacking problem has increased the paper's impact, and it is an exciting experiment with quite some distinctions from other photo-induced metastable states.

We would like to express our great appreciation to all three referees for their 2nd round reviews of our manuscript and recommendations. Below is a list of our detailed response to their new reports.

Reviewer #1 (Remarks to the Author):

The revised manuscript by Liu et al is an improvement on the initial submission. I still feel that it could be improved. In particular, although comments on the literature I suggested have been added, it feels it has been done more to satisfy me, rather than to improve the paper. i.e. the papers have not had an real influence on the general discussion in the paper. If the authors do not think these papers add anything interesting to the paper they do not need to be added. I agree with the authors that this appears to be more complicated than simple stacking faults, but is that complication really important? Perhaps that is the case in TaS₂ too, it has not been explicitly measured. Such things could be commented on.

-----response-----

We thank the referee for the comments. For the issue of 'stacking fault', as our comment for the previous round reviewer: 'We mentioned that the main change from the TEM image appears to be a Ta lattice shear distortion along a-axis in the photoexcited LR phase. It is not really a stacking fault as observed in 1T-TaS₂ after photoexcitation, therefore we refrain from using the term "stacking fault"'. Maybe, stacking fault could happen in other conditions like high pressure, high external strain, or other ways of illuminating. However, for our present work, the certain in-chain displacement of Ta-Ta atoms in Ta₂NiSe₅ in the photoinduced LR state indicated that the distortion appears more complicated than a simple stacking fault scenario.

Ultimately, I am a little unsure of if the authors are trying to communicate about this phase transition. On the one hand, the results are clear and almost speak for themselves, especially with regards to the excitonic insulator. In particular, I do not know if the authors are claiming this process is driven by a particular phonon or peak field, or something else.

-----response-----

The energy of 800 nm pulses is far larger than the possible phonon frequencies in Ta₂NiSe₅. So the process should not be a particular phonon matter. As we have discussed in previous round reviewer by comparing our results with the literatures', the process is most likely driven by the peak field.

That said, although I would encourage the authors to try to focus the argument a little more, I would also be happy to accept it in the current form. However, in any case, the paper needs to be checked as I found several mistakes of which only a few are highlighted here:

Ultrashort laser pulse not only is a powerful -> Ultrashort laser pulses are not only a

powerful...

The samples, in the thickness of -> The samples, with thicknesses of

Two of them (2.0- and 4.0- THz) got disappeared -> two of them (2.0- and 4.0- THz) disappeared

Is EI defined? It is mentioned twice but I cannot see a definition

-----response-----

We thank the referee for the recommendation and comments. We went through the manuscript and made corrections.

Reviewer #2 (Remarks to the Author):

Dear editor, dear authors,

thanks for getting back to me with the revised manuscript. As said in the previous round of reports the finding of the phase switching effect alone I do not judge as exciting enough to recommend publication in Nat. Comm.

Also in the new manuscript there is no additional information on the nature of the phase transition that would prove a novel type of transition.

However, as mentioned I do find the material system Ta₂NiSe₅ highly interesting and the nature of structural phase transition in that material is under hot discussions both both from experimental and theoretical points of view.

In the new manuscript the authors link to the suggested additional pre-prints and papers I did suggest and discuss their own results in this context. As such I feel that the paper adds important information to a question of immediate interest in the community on a very decisive question on the phase transition in Ta₂NiSe₅.

Therefore I do recommend publication of the present manuscript, now.

-----response-----

We thank the referee for the recommendation.

I also did read through the responses to the other reports and find the answers of the author sufficient. Also the answers the open points and "rejections" to requests on additional data that would be nice to have but that go clearly beyond the scope of the present study or are not easily gained experimentally is done well and justified. The latter also counts for requests made by myself.

Some minor points that may or may not be helpful:

(1) to the report of reviewer 1:

In the question of presenting figure 3e/h. I would not mirror the image. The red line in the data made it clear how to read the figure - at least for me. Maybe as a compromise one could align the red line in both figures to the same position and show a slightly larger cut in the figures.

-----response-----

We thank the referee for the comment. We re-check figure 3e/h and find no ambiguity. As the reviewer also commented 'The red line in the data made it clear how to read the figure - at least for me.', the figure 3e/h was not further modified.

(2) To my point on the discussion of the TEM figure. I am fine with the response they give to me in the letter. Maybe the authors could also add this statement to the main text.

In their revised manuscript they just write "the above coherent phonon excitation"....

In detail there is an offset displacement that is in the direction of the phonon mode, the oscillatory motion would be then on top of that.

-----response-----

We thank the referee for the comment. We added the following sentence in the manuscript: 'It is expected that an offset displacement gradually develops in the direction of the in-chain phonon mode, and the oscillatory motion would be on top of that. Increasing the pump field would lead to a much larger offset displacement, similar to what has been observed in WTe₂ and MoTe₂ [33,34]. Above a threshold pump field, the lattice would not recover to its original state and a new equilibrium state is reached eventually.'

(3) On the point of the discussion of the exciton binding energy. I do understand the points of the authors. However that is what one would theoretically expect as a scenario. But there is no experimental evidence for the exciton binding energy/exciton radius.

Maybe instead of "apparently, the ex binding energy must be...." the others could tone down a bit to "The absence of a phase transition suggests or is compatible with...."

-----response-----

We thank the referee for the comment. We made change accordingly in the main text: "The absence of a phase transition suggests very small exciton binding energy in this phase."

Reviewer #3 (Remarks to the Author):

Authors have made connections to Raman, and the stacking in 1T-TaS₂ problem and

appropriately modify/improve the main text.

I can imagine that the measurement of less trivial transport properties, like the Hall conductivity or the thermopower, is challenging for authors. However, I would expect a more proactive approach to analyzing the pulse duration as it would provide additional insights into the electronic-lattice interplay. The lack of the latter is still my main criticism of the paper.

I like the summary table on different transient experiments that authors have provided in the reply. I believe it would be useful to include it in the SM and point a reader to it due to the "confusion in the field".

-----response-----

We thank the referee for the comment. We have added the summary table as Table. S1 in SM.

A side remark. I've rechecked reference 23 [Mor at.al], and your classification could be a bit misleading [or the text in their paper]. Indeed, authors measure the saturation effect, but on Fig.3 we see that on the timescale longer than ~ 1 ps the peak position shrinks below the equilibrium value~(at least a partial gap closure) and the EDC data in the SM [Fig S2] shows a strong signal at the chemical potential. Unfortunately, authors do not show high-excitation and long time data, but I strongly suspect that they look semimetallic-like. I think the discussion in conclusion ["above the saturation fluence the band energy gaps widened, reflecting an increase of the exciton condensate density in the EI phase in Ta₂NiSe₅ [23] etc."] is still fine, but you could comment on the issue.

-----response-----

Because we can't get more detailed experimental information from the reference 23 [Mor at.al], we refrain from making further comment on that issue.

I'd also suggest that authors mention in the main text that any polarization can induce PI state as this is a nontrivial effect.

-----response-----

We find the effect can be induced by pump pulse polarization not parallel to the a-axis. However, we did not make a careful calibration whether the effect is the same as the polarization parallel to the a-axis. For this reason, we refrain from adding such a claim on the manuscript.

If I sum up, I would recommend the publication as the connection with the Raman and the stacking problem has increased the paper's impact, and it is an exciting experiment with quite some distinctions from other photo-induced metastable states.

-----response-----

We thank the referee again for the recommendation.

List of major changes:

1. The paper have been rechecked for several typos.
2. The table of pump laser parameters from different references in the main text is provided as Supplementary Table. 1 in SM.